# Impacts Analysis of Alien Macroinvertebrate Species in the Hydrographic System of a Subalpine Lake on the Italian–Swiss Border

Daniele Paganelli [ID], Lyudmila Kamburska [ID], Silvia Zaupa, Laura Garzoli [ID] and Angela Boggero *[ID]

National Research Council-Water Research Institute (CNR-IRSA), Corso Tonolli 50, 28922 Verbania, Italy; daniele.paganelli@irsa.cnr.it (D.P.); lyudmila.kamburska@irsa.cnr.it (L.K.); silvia.zaupa@irsa.cnr.it (S.Z.); laura.garzoli@irsa.cnr.it (L.G.)
* Correspondence: angela.boggero@irsa.cnr.it

**Abstract:** The potential invasiveness of alien macroinvertebrate species in the Italian/Swiss hydrographic system of Lake Maggiore (NW Italy) was assessed through the Aquatic Species Invasiveness Screening Kit, a risk assessment tool developed for quantifying the impacts of alien species on the commercial, environmental, and species traits sectors. Data were collected using the databases provided by two regional environmental agencies in northern Italy (Lombardy and Piedmont regions) and by the governmental monitoring program of Switzerland, which were integrated with a systematic literature search on Google scholar and ISI Web of Science. In the assessment area, 16 macroinvertebrate invasive alien species were reported: nine mollusks, four decapods, and three amphipods. The species assessment indicated seven species with a high level of invasiveness: *Procambarus clarkii*, *Faxonius limosus* (formerly, *Orconectes limosus*) and *Pacifastacus leniusculus*, *Dreissena polymorpha*, *Corbicula fluminea*, *Sinanodonta woodiana*, and *Pseudosuccinea columella*. The results allow invasive species managers to understand which species to focus their monitoring on in the near future in order to track IAS movements and limit their spread within the hydrographic system and to provide the identification and refinement of concerted bilateral strategies aimed at limiting the impacts of these species. They also account for the implications of future climate change on the invasion potential of each species.

**Keywords:** freshwater; invasiveness assessment; AS-ISK; non-native macrobenthic fauna; Lake Maggiore; northern Italy; Switzerland

## 1. Introduction

Invasive alien species (IAS) affect all ecosystems around the world, and they are recognized as a serious threat to biodiversity. Their increasing number worldwide is strongly linked to international trade, which represents the most significant component among human-mediated introductions [1].

The ecological, economic, and social impacts of IAS are well known, and this is especially true for those that cause direct or indirect problems to human beings [2–4]. As a result, IAS are the subject of several international agreements (i.e., the Convention on Biological Diversity, the International Plant Protection Convention, and the United Nations Sustainable Development Goals), initiatives (i.e., the Intergovernmental Science-Policy Platform on Biodiversity and Ecosystem Services), regulations (i.e., EU Regulation 1143/2014), and conservation strategies (EU Biodiversity Strategy to 2020, European Commission, 2011) that aim to tackle biodiversity loss by preventing the spread of IAS, prioritizing surveillance, early eradication, and long-term control.

To address the problem of biological invasions effectively, it is important to understand which biological and ecological traits of alien species may favor the colonization of a new environment. It is also important to analyze factors such as invasion history, the possible

effects of climate change, the range of possible sources of introduction, the status of species or habitats under threat that affect the introduction, and the establishment and spread of invasive species into newly colonized environments. The combination of these analyses identifies potential risks at an early stage and suggests justified measures to mitigate their impacts [5].

Furthermore, species invasion, establishment, spread, and impacts are also influenced by the abiotic factors of the receiving environment; thus, another approach to study invasive species is to define the environmental vulnerability of an area to the biological invasion of a single or a group of species [6].

One way to predict and prevent the introduction of IAS is to study the invasiveness records of these species in different areas, especially in neighboring areas. In this context, it is pivotal to assess their risk of entering new environments [7–9], overcoming national borders and not limiting surveys and monitoring efforts to specified geographical or national borders [10].

However, in general, the first step for defining the risk level of an invasion of a species is to perform a risk assessment of the invaders [11]. A risk assessment is generally carried out to (a) evaluate the likelihood of the introduction of invasive species in natural ecosystems, (b) gain insights into the main pathways and conveyances prior to establishment to guide decisions on prevention measures, and (c) support decision making concerning the allocation of resources for the monitoring, control, and/or eradication of established populations, including common actions between countries to mitigate impacts on the environment or human health.

In Europe, there are about 12,000 alien species, but fewer than 15% of these have become IAS (European Union regulation 1143/2014). Among these, EU regulation 1143/2014 has defined 67 species as Species of Union Concern: 36 plants, 12 mammals, 5 birds, 4 fish, 1 amphibian, 1 reptile, and 8 invertebrates.

In Italy, according to a recent assessment, there are about 3000 alien species, 150 of which are freshwater species (Tricarico, personal communication), and only 4 of which are macroinvertebrates from the EU List of Union Concern, namely *Orconectes limosus* (which went under reclassification in August 2017, with the genus changing to *Faxonius* [12]), *Pacifastacus leniusculus*, *Procambarus clarkii*, and *Procambarus virginalis* [13].

Nonetheless, the invasion of freshwater ecosystems is recognized as a problem of global concern, and the management of aquatic IAS is also a difficult challenge [14–16] because the highly interconnected hydrographic system of inland waters acts as a corridor for their spread. Furthermore, IAS show high ecological adaptation, and some of them have already invaded freshwater ecosystems, becoming an important part of freshwater communities, both in terms of the number of species and biomass, leading to biodiversity loss and environmental homogenization [17,18].

The risk assessment of species that, for various reasons, are not deeply studied or that were introduced in the past and are now well established, is not always easy. Although freshwater aquatic invertebrates are one of the most successful invader groups of organisms due to their biological traits and due to their ability to colonize both lentic and lotic ecosystems [19], their impacts are often harder to assess compared to larger species. Apart from species such as *Procambarus clarkii* [20] or *Dreissena polymorpha* [21], many macrobenthic species are not easy to identify morphologically, or their economic impact is less studied [22,23].

Therefore, the aim of this paper is to analyze the potential invasiveness of the macroinvertebrate alien species detected in the hydrographic system of Lake Maggiore, one of the large, deep lakes on the border that northern Italy shares with Switzerland. We focused our assessment on alien macroinvertebrate species because they are increasingly abundant in European freshwaters compared to other taxonomic groups [24,25].

The impacts of IAS will be described by means of a specific risk assessment tool called the Aquatic Species Invasiveness Screening Kit (AS-ISK) [26] (see methods for details), which is applicable to all aquatic organisms from freshwater vertebrates [27] to marine [28]

and freshwater invertebrates [29]. We used this tool to assess macroinvertebrate alien species, defining the potential biological, environmental, and socio-economic problems that they can cause in the major tributaries of the hydrographic system of Lake Maggiore.

## 2. Materials and Methods

### 2.1. Study Area

The Lake Maggiore hydrographic system covers a total surface area of 6.599 km$^2$, an area that is roughly divided in half between the Italian and Swiss territory. It is crossed by a complex hydrographic network that is composed of approximately 35 tributaries of different sizes and importance, and one main outlet, the River Ticino. Within the catchment area, water is extensively used for hydroelectric power production, agriculture, and industry. Moreover, some areas and their alpine surroundings also provide other ecosystem services (i.e., tourism) that support the economy of the most densely populated and productive area in Italy [30].

For this study, we considered data from an assessment area (hereafter called Risk Assessment Area, RA area) that consisted of the major tributaries of Lake Maggiore, namely the rivers Ticino inlet, Maggia and Verzasca (Switzerland), the rivers Ticino outlet, Toce and Tresa (Italy), and some of the minor tributaries in the Italian territory (the rivers Cannobino, San Bernardino, San Giovanni, Giona, Strona, Bardello, Margorabbia, Boesio, Vevera, and Erno) (Figure 1).

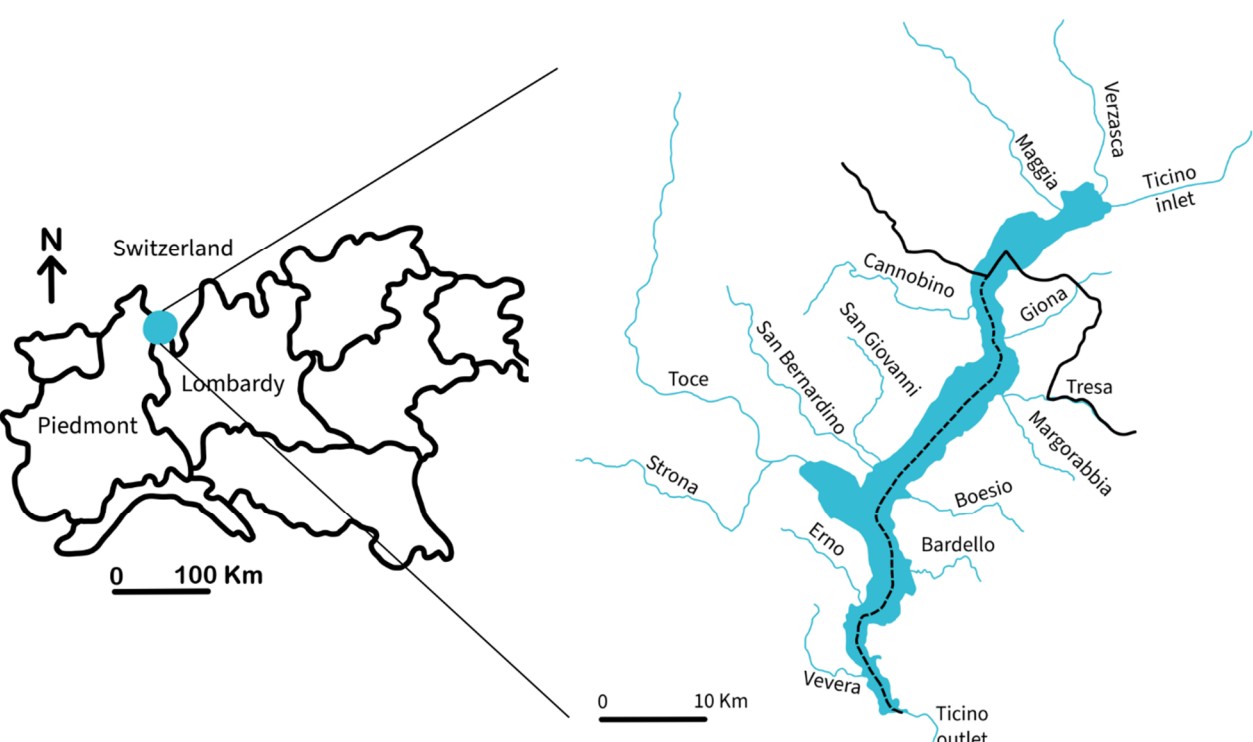

**Figure 1.** Study area: Lake Maggiore and its main tributaries. Dotted line: border between Lombardy and Piedmont regions; solid line: border between Italy and Switzerland.

### 2.2. Data Collection

The alien species database used for our purposes includes data from regional monitoring programs from the Environmental Agencies in the Piedmont (ARPA Piemonte) and Lombardy (ARPA Lombardia) regions in addition to the data from the governmental monitoring program of Switzerland and records from literature references.

Information from the scientific literature was sought via Google Scholar and Web of Science using search queries such as "alien macrobenthos species" or "alien macroinverte-

brates species" or "freshwater alien species" followed by the name of each area and river. Moreover, we used the same search engines to collect ecological information on each of the detected aquatic alien species.

Presence-only data consisting of locations with known presences were used, and our research was limited to only considering the impacts of aquatic alien species independently of their relative abundance, which may vary considerably across the distribution range of a species.

*2.3. Data Processing*

The detected alien species were screened by means of the Aquatic Species Invasiveness Screening Kit (AS-ISK v2.3), a multilingual screening tool that has been translated into 32 languages and that is available for free download at www.cefas.co.uk/nns/tools/ (accessed on 5 March 2021) [26,31].

The AS-ISK tool is composed of 49 questions that examine the impact of an alien species from a biogeographical, biological, and economic point of view. The combination of these three types of impact produces a Basic Risk Assessment (BRA) score. A further six questions ask the user to define the Climate Change Assessment (CCA) of the species, describing the potential effects of future climate change on the risk of the species' introduction, establishment, dispersal, and socio-economic impacts. Moreover, for each question, a brief guide with the explanation of its aim is provided.

The final BRA and BRA + CCA scores vary from −20 to 68 and from −32 to 80, respectively, and the combination of the BRA with the CCA scores gives the species final level of invasiveness.

AS-ISK scores higher than 1 indicate an IAS that may have a medium or high risk of invasiveness. On the contrary, species with scores < 1 are not considered to be invasive in the RA area. The boundaries between a medium and high level of invasiveness depends on the threshold value that is used. When using a small dataset, it is possible to borrow these values from other studies [32,33].

In the present case, due to the relatively small number of detected alien species, it was not possible to calibrate the necessary threshold for the AS-ISK scores; therefore, we decided to apply the same values (BRA = 36 and BRA + CCA = 38) used by Paganelli et al. [18] for the macrobenthic alien community in the secondary hydrographic system of the River Ticino.

Furthermore, in order to achieve a valid risk assessment, the user is requested to provide a confidence level to the answer as per the Intergovernmental Panel on Climate Change (1 = low; 2 = medium; 3 = high; 4 = very high), and to justify each answer, a detailed reference list (grey or scientific) is requested [26].

In order to appraise the final invasiveness level of each species obtained through the BRA and BRA + CCA scores, each species was preventively classified as invasive or non-invasive following a four steps approach. Firstly, an online search of any literature reference reporting threats to humans/anthropogenic activities was performed, and the information obtained through that search allowed us to define each species as being harmless or as a potential pest. Secondly, a search was performed using Web of Science and Google Scholar to check whether there were any references in literature regarding the impacts of the species on invaded ecosystems. Thirdly, the EU lists of IAS of Union Concern were searched, and lastly, the Global Invasive Species Database (GISD—iucngisd.org) was used to discover whether each species was listed as invasive in the database managed by the Invasive Species Specialist Group (ISSG) of the International Union for Conservation of Nature (IUCN) Species Survival Commission.

## 3. Results

In total, in the RA area, 16 macroinvertebrate aquatic alien species were reported: nine mollusks, four decapods, and three amphipods. According to the Piedmont Environmental Agency (ARPA Piemonte), two IAS are present in the tributaries of Lake Maggiore, and

the Lombard Environmental Agency (ARPA Lombardia) indicates the presence of 10 IAS, while in Switzerland, the governmental monitoring program indicates the presence of 9 IAS in the tributaries of the lake (Table 1).

**Table 1.** List of freshwater alien species examined with AS-ISK in the catchment area of Lake Maggiore. The X indicates the presence of the species in the database.

| Scheme | Common Name | Lombardy Region | Piedmont Region | Switzerland |
|---|---|---|---|---|
| Bivalves | | | | |
| *Corbicula fluminea* (Müller, 1774) | Asiatic clam | X | X | - |
| *Dreissena polymorpha* (Pallas, 1771) | Zebra mussel | X | X | - |
| *Sinanodonta woodiana* (Lea, 1834) | Chinese pond mussel | X | - | - |
| Gastropods | | | | |
| *Ferrissia californica* (Rowell, 1863) | \ | X | - | X |
| *Gyraulus parvus* (Say, 1817) | Ash gyro | - | - | X |
| *Physella acuta* (Draparnaud, 1805) | Bladder snail | X | - | X |
| *Planorbis corneus* (Linneaus, 1758) | Great ramshorn | - | - | X |
| *Potamopyrgus antipodarum* (Gray, 1843) | New Zealand mud snail | X | - | X |
| *Pseudosuccinea columella* (Say, 1817) | American ribbed fluke snail | X | - | - |
| Decapods | | | | |
| *Faxonius limosus* (Rafinesque, 1817) | Spiny cheek crayfish | X | - | - |
| *Pacifastacus leniusculus* (Dana, 1852) | Signal crayfish | - | - | X |
| *Pontastacus leptodactylus* (Eschscholtz, 1823) | Turkish crayfish | - | - | X |
| *Procambarus clarkii* (Girard, 1852) | Red swamp crayfish | X | - | - |
| Amphipods | | | | |
| *Gammarus roeselii* Gervais, 1835 | \ | X | - | - |
| *Cryptorchestia cavimana* (Heller, 1865) | \ | - | - | X |
| *Synurella ambulans* (Müller, 1846) | \ | - | - | X |

The preventive analysis of the invasiveness level of the aquatic alien species highlighted that 6 species out of 16 have a direct or indirect impact on human beings or on their activities, while 8 out of 16 species have a demonstrated impact on the local biodiversity. Moreover, this analysis showed that almost none of the alien macrobenthic species are included in the EU list of IAS of Union Concern or are even listed in the IUCN database (Table 2).

**Table 2.** Preventive analysis of the invasiveness level of the aquatic alien species listed in the catchment area of Lake Maggiore. N = no impacts, Y = with impacts. In brackets, we reported the literature collected about the impacts of the alien species.

| Species | 1st Step (Threats to Humans) | 2nd Step (Impacts on Native Biodiversity) | 3rd Step (List of Union Concern) | 4th Step (Listed as Invasive in the IUCN Database) |
|---|---|---|---|---|
| Bivalves | | | | |
| *Corbicula fluminea* | Y [34] | Y [34] | N | Y |
| *Dreissena polymorpha* | Y [35] | Y [36] | N | Y |
| *Sinanodonta woodiana* | N | Y [37] | N | N |

**Table 2.** *Cont.*

| Species | 1st Step (Threats to Humans) | 2nd Step (Impacts on Native Biodiversity) | 3rd Step (List of Union Concern) | 4th Step (Listed as Invasive in the IUCN Database) |
|---|---|---|---|---|
| Gastropods | | | | |
| *Ferrissia californica* | N | N | N | N |
| *Gyraulus parvus* | N | N | N | N |
| *Physella acuta* | N | Y [38,39] | N | N |
| *Planorbis corneus* | N | N | N | N |
| *Potamopyrgus antipodarum* | N | Y [40] | N | Y |
| *Pseudosuccinea columella* | Y [41] | N | N | N |
| Decapods | | | | |
| *Faxonius limosus* | Y [42] | Y [43] | Y | N |
| *Pacifastacus leniusculus* | Y [42] | Y [44] | Y | Y |
| *Pontastacus leptodactylus* | N | N | N | N |
| *Procambarus clarkii* | Y [45] | Y [46] | Y | Y |
| Amphipods | | | | |
| *Gammarus roeselii* | N | N | N | N |
| *Cryptorchestia cavimana* | N | N | N | N |
| *Synurella ambulans* | N | N | N | N |

Considering the 880 resulting responses (i.e., 55 questions × 16 alien species), only 43 (4.8% of the total) were Not Available (N.A.), and the level of confidence varied from 0.69 to 0.88 and from 0.65 to 0.90, respectively, for the BRA and BRA + CCA scores.

The BRA scores ranged from 46.5 to 8 and, according to the threshold imposed for the BRA, 7 species out of 16 were classified as having a high level of invasiveness, while the remaining 9 species were classified as having a medium level of invasiveness.

A similar ranking of alien species was obtained using the BRA + CCA: the scores ranged from 58.5 to 4.5. In this case, eight species were classified as having a high risk of invasiveness, and the remaining eight were classified as having a medium level of invasiveness (Figure 2).

The assessment results highlighted that the group of species with the highest level of invasiveness (BRA scores) was composed of three decapods, three bivalves, and one gastropod, namely *Procambarus clarkii* (BRA = 46.5; BRA + CCA = 58.5), *Faxonius limosus* (BRA = 43.5; BRA + CCA = 55.5), *Pacifastacus leniusculus* (BRA = 39.5; BRA + CCA = 51.5), *Dreissena polymorpha* (BRA = 39.5; BRA + CCA = 51.5), *Corbicula fluminea* (BRA = 39.5; BRA + CCA = 51.5), *Sinanodonta woodiana* (BRA = 38.5; BRA + CCA = 48.5), and *Pseudosuccinea columella* (BRA = 36; BRA + CCA = 40).

In contrast, the species with the lowest scores were two amphipods, namely *Synurella ambulans* (BRA = 8; BRA + CCA = 8) and *Chryptorchestia cavimana* (BRA = 8.5; BRA + CCA = 4.5) (Figure 2).

Considering the three sectors (commercial, environmental and species traits) that the AS-ISK results highlight, the species with the highest impact on the commercial and environmental sectors was *P. clarkii,* with scores of 18 and 16, respectively. Regarding the species traits sector, *C. fluminea* showed the highest score (36), followed by *D. polymorpha,* with a score of 32 (Table 3).

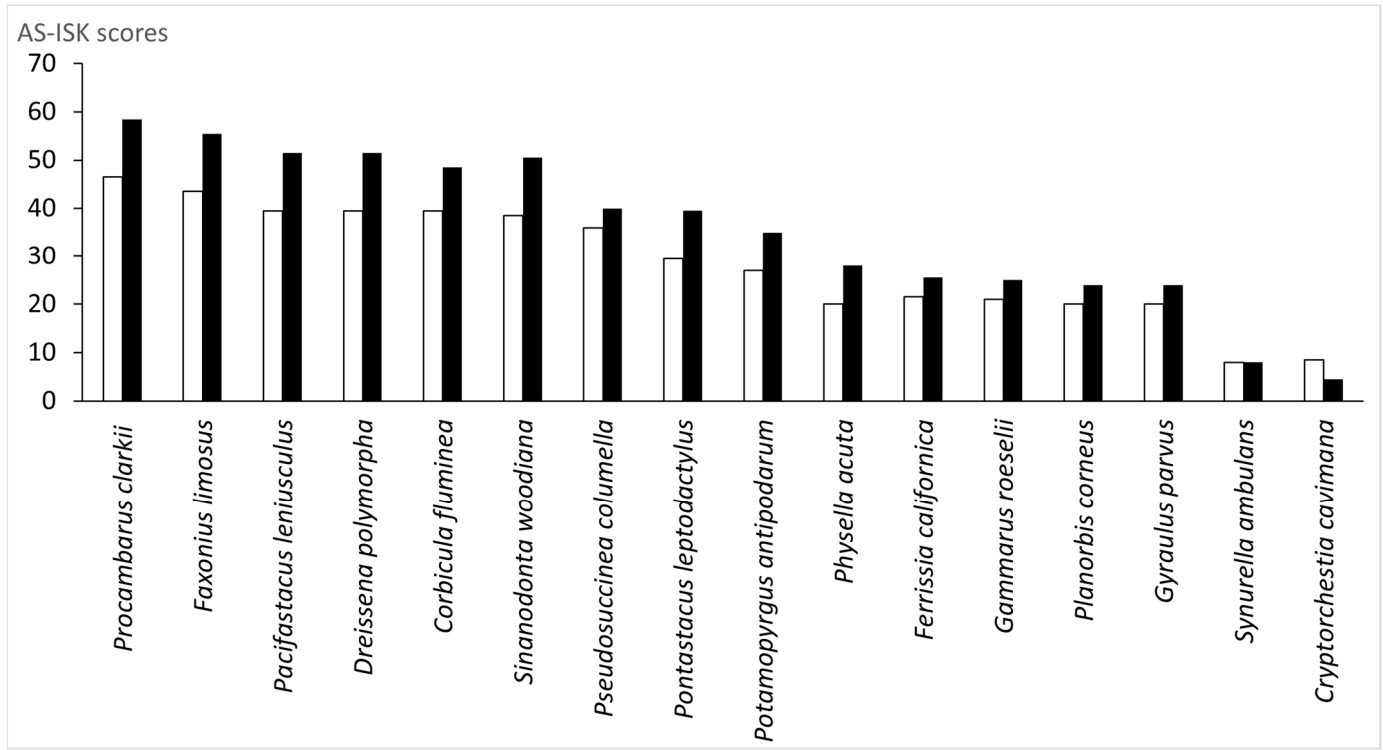

**Figure 2.** Evaluation of the potential level of invasiveness of the alien species in the assessment area. White columns: Basic Risk Assessment score (BRA); black columns: Basic Risk Assessment and Climate Change Assessment score (BRA + CCA).

**Table 3.** AS-ISK results. AS-ISK sector affected and score partition and level of confidence for each alien species. +: low level of risk; ++: medium level of risk; +++: high level of risk; ++++: very high level of risk. Scenario I uses the thresholds proposed by Paganelli et al. [18], and scenario II uses the thresholds proposed by Vilizzi et al. [47].

| | AS-ISK Sector Affected | | | Score Partition | | | Level of Risk | | Level of Confidence | |
| --- | --- | --- | --- | --- | --- | --- | --- | --- | --- | --- |
| IAS | Commercial | Environmental | Species Traits | Biogeography/ Historical | Biology/ Ecology | Climate Change | BRA | BRA + CCA | BRA | BRA + CCA |
| *Procambarus clarkii* | 18 | 16 | 30 | 19.5 | 27 | 12 | +++ | +++ | 0.88 | 0.90 |
| *Faxonius limosus* | 15 | 16 | 30 | 16.5 | 27 | 12 | +++ | +++ | 0.82 | 0.83 |
| *Pacifastacus leniusculus* | 15 | 13 | 29 | 13.5 | 26 | 12 | +++ | +++ | 0.74 | 0.73 |
| *Dreissena polymorpha* | 14 | 11 | 32 | 15.5 | 24 | 12 | +++ | +++ | 0.87 | 0.86 |
| *Corbicula fluminea* | 10 | 10 | 36 | 11.5 | 28 | 12 | +++ | +++ | 0.75 | 0.75 |
| *Sinanodonta woodiana* | 15 | 11 | 30 | 16.5 | 22 | 10 | +++ | +++ | 0.74 | 0.74 |
| *Pseudosuccinea columella* | 10 | 7 | 31 | 13 | 23 | 4 | +++ | +++ | 0.69 | 0.65 |
| *Pontastacus leptodactylus* | 12 | 12 | 21 | 8.5 | 21 | 10 | ++ | +++ | 0.81 | 0.80 |
| *Potamopyrgus antipodarum* | 3 | 8 | 29 | 2 | 25 | 8 | ++ | ++ | 0.80 | 0.77 |
| *Physella acuta* | 2 | 5 | 25 | 1 | 19 | 8 | ++ | ++ | 0.83 | 0.82 |
| *Ferrissia californica* | 2 | 1 | 27 | 2.5 | 19 | 4 | ++ | ++ | 0.76 | 0.76 |
| *Gammarus roeselii* | 3 | 2 | 25 | 4 | 17 | 4 | ++ | ++ | 0.72 | 0.71 |

**Table 3.** *Cont.*

| IAS | AS-ISK Sector Affected | | Score Partition | | | | Level of Risk | | Level of Confidence | |
|---|---|---|---|---|---|---|---|---|---|---|
| | Commercial | Environmental | Species Traits | Biogeography/ Historical | Biology/ Ecology | Climate Change | BRA | BRA + CCA | BRA | BRA + CCA |
| *Planorbis corneus* | 2 | 1 | 25 | 4 | 16 | 4 | ++ | ++ | 0.73 | 0.72 |
| *Gyraulus parvus* | 2 | 1 | 25 | 4 | 16 | 4 | ++ | ++ | 0.73 | 0.72 |
| *Synurella ambulans* | 3 | 1 | 9 | 2 | 6 | 0 | ++ | ++ | 0.73 | 0.71 |
| *Cryptorchestia cavimana* | 3 | 1 | 6 | 1.5 | 7 | −4 | ++ | ++ | 0.71 | 0.70 |

A detailed analysis of the AS-ISK score partition revealed that *P. clarkii* and *F. limosus* shared the same scores in the biology/ecology (27) and climate change (12) sectors, while they showed a slight difference in the biogeography sector (*P. clarkii*, 19.5 and *F. limosus*, 16.5). Despite obtaining the same score in the climate change sector, *P. leniusculus* obtained a lower overall level of invasiveness compared to the other two crayfish species due to its lower scores in the biology/ecology (26) and the biogeography sectors (13.5) (Table 3).

Excluding *D. polymorpha* and *C. fluminea*, which obtained almost the same scores as all of the decapods in all of the sectors (and sometimes higher scores, e.g., *Corbicula fluminea* in the bio-ecology sector), bivalves generally obtained lower scores than decapods in all three sectors explored by the AS-ISK. Finally, in the biogeography sector, *S. woodiana* was the bivalve with the highest score (16.5) (Table 3).

At the opposite end of the invasiveness chart, two amphipod species (*S. ambulans* and *C. cavimana*) obtained the lowest scores for the entire dataset in all of the investigated sectors (Table 3).

## 4. Discussion

Environmental risk assessment tools are methods for estimating the probability and severity of an undesired event (e.g., biological invasions) and try to quantify and evaluate all of the factors that have an impact on the assessed environment. In contrast, a preventive risk analysis indicates whether an event is likely to be undesired according to the information collected in the literature. When the two analyses agree, it indicates that the assessment tool is able to predict the potential risks.

In our case study, the preventive risk analysis indicated that about half of the total species that were assessed are threats to human beings and activities or to the native biodiversity. These species were decapods (i.e., crayfish) and mollusks: *Procambarus clarkii, Faxonius limosus, Dreissena polymorpha, Corbicula fluminea, Pseudosuccinea columella, Pacifastacus leniusculus, Physella acuta, Potamopyrgus antipodarum,* and *Sinanodonta woodiana*.

However, although almost 50% of the assessed alien species have impacts on human activities or the biodiversity of the area, only five of them are on the IUCN database of the worst alien species; moreover, only three of them are on the EU list of Union Concern (Table 2).

Not surprisingly, almost all of the crayfish species present in the hydrographic system of Lake Maggiore are classified as having a "high risk" of invasiveness. Generally speaking, crayfish are among the most renowned invasive aquatic species in freshwater ecosystems. They are active burrowers, and their intense burrowing activity can increase water turbidity [48]. Moreover, the burrowing action of crayfish induces the extensive perforation of canal banks and their rapid collapse, often producing damage to agricultural fields [49]. In addition, crayfish can impact ecosystem services by preying on the fish fauna and by destroying the aquatic vegetation [50–52]. Another ecological impact of American invasive crayfish is their direct contribution to the disappearance of the Italian native crayfish *Austropotamobius pallipes* because the former acts as a vector of the crayfish plague *Aphanomyces astaci* [53–55].

The species with the highest level of invasiveness was *Procambarus clarkii,* and several studies have already been performed on the impact of this species at various levels [52].

*Procambarus clarkii* is known as one of the most invasive species in the world due to its high ecological and economic impacts; thus, information on this IAS is more accurate than it is for other species. Consequently, the level of confidence for the AS-ISK answers was quite high. One of the reasons why *P. clarkii* started its successful expansion in the wild is because of its opportunistic behavior and physiology. Furthermore, it is very tolerant and adaptable to a wide range of aquatic conditions, including moderate salinity, low oxygen levels, extreme temperatures, and pollution [55]. Its high rate of reproduction and high dispersion capacity through more than one vector (active dispersion, birds, and anthropogenic vectors), increase its ability to colonize rivers, wetlands, ponds, and almost all types of freshwater habitats [55–57].

The introduction of this species in Europe and other parts of the world was due to its economic value as a food source: since the 18th century in North America, crayfish culture has been very common, especially in Louisiana [58]. In the past, this culture also became quite common in Europe, but nowadays, the economic value of this industry is not relevant anymore.

Among the other three crayfish species on the list, only *Faxonius limosus* reached a level of invasiveness that was comparable to that of *P. clarkii*, while *Pacifastacus leniusculus* reached a slightly lower level of invasiveness (Figure 2). The similarity between these three species is due to the same level of impact that they have on the commercial, environmental, and biological sectors and their similar biogeographic and ecological characteristics (Table 3).

On the contrary, *Pontastacus leptodactylus* was ranked with a lower level of invasiveness (BRA score) because it is less adaptable compared to the other crayfish varieties (Figure 2). It tolerates a lower level of anthropogenic impact, and, above all, it can be affected by crayfish plague [59]. Consequently, this species has lower scores in the three sectors assessed by the AS-ISK tool (Table 3).

Another group of invasive species that emerged from the preventive analysis is mollusks, mainly bivalves. Although they are known for their "sluggishness" and time-consuming movements that take place "at a snail's pace", some of them have become perfect examples of invasive species due to their rapid spread [60]. Invasive bivalves have different types of impacts on human activities; for example, when abundant, they can damage industrial pipes or can impact tourism [61].

Invasive crayfish and bivalves share a negative characteristic: they have a high impact on the native species, causing a decline in local biodiversity and, consequently, a general biotic homogenization of freshwater communities [62].

The AS-ISK results indicated two mollusks with a high level of invasiveness: *Corbicula fluminea* and *Dreissena polymorpha* (Figure 2). Both have the typical traits of successful invaders, and in the assessment area, they are very well established, reaching high densities [18,63–65].

Despite being relatively small in size, *Corbicula fluminea* is considered to be one of 100 of the world's worst invasive alien species (DAISIE—Delivering Alien Invasive Species Inventories for Europe—http://www.europe-aliens.org (accessed on 1 June 2021), and it has a variety of impacts, especially on an ecological level (Global Invasive Species Database, GISD—iucngisd.org) [66,67]. In the late 1990s, this IAS was reported in Italy for the first time [68], and since then, it has spread to almost all freshwater environments in Northern Italy [69]. The high invasiveness of *C. fluminea* is due to its tolerance to the main environmental variables (i.e., dissolved oxygen, water temperature, and nitrogen concentration) but also to its high spreading potential [34,70,71] and its high competition for space and resources with native species [72,73].

*Dreissena polymorpha* is considered one of the most successful freshwater invaders worldwide [74], and its spread was strictly related to human activities: aquarium trade, ballast water, and inland navigation were the main pathways of introduction [75]. Adult

specimens attach to boat hulls and to any hard substrates with their byssal threads, so they can cover long distances, even upstream [76]. For example, in Italy, they were first reported in 1971 in Lake Garda immediately after a leisure nautical event [77]. The species is an ecosystem engineer, and its presence causes the alteration of the trophic web [73]. Adults can filter up to two liters of water each day, reducing the amount and biomass of phytoplankton, thus increasing water clarity but limiting the food for fish larvae or other freshwater mussels [78]. Moreover, high densities of *D. polymorpha* represent another stress for the native mussels, which could literally be suffocated by its presence [79]. It may also bioaccumulate pollutants, which can poison other animals in the food web (DAISIE, 2006). Nevertheless, its presence is sometimes positive: for example, it favors macrobenthic fauna density and taxonomic richness by increasing the bio-deposition of organic waste and food resources, and it can provide greater habitat complexity [80].

All of the other alien species assessed in this study were described by the preventive analysis as not being particularly harmful to humans or as not posing a particular threat to the biodiversity, and the AS-ISK classified them as having a medium risk of invasiveness.

This result could be due to several reasons: for example, some macrobenthic species are often under-studied, and information on their auto-ecology is difficult to find; consequently, it is more difficult to define their impacts (e.g., *Ferrissia californica*, formerly known as *F. wautieri* or *F. fragilis* or *F. clessiniana*, see [81]), as testified by the lower level of confidence obtained for the final assessment results (Table 3).

On the other hand, some species do not show the typical invasive biological traits, and therefore, their presence is limited to specific habitats, or they do not show economic impacts on human activities (e.g., the amphipods *Synurella ambulans* or *Cryptorchestia cavimana*).

Furthermore, some species have been in the RA area for a long time, such as *Physella acuta* (first record in Italy in 1866) or, more recently, *Potamopyrgus antipodarum*, which was first recorded in Italy in 1961 [82]; as such, they could almost be considered to be part of the "new" local macrobenthic community.

Finally, the last six questions of the AS-ISK assessed the effects of climate change on the examined species. To answer this group of questions, it is more advisable to take advantage of the information from existing climate change research to indicate how future climate change could influence the impact of alien species in the RA area. When this is not possible, the tool suggests using "expert judgment". Thus, the answers and their level of confidence depend on how long those species have been studied and how strong their impact on ecosystems or on human activities is.

In our assessment, for less-studied alien species (i.e., amphipods), this section was evaluated using "expert judgment" because it was not possible to find specific studies on this topic, whereas for the most-studied ones (i.e., decapods), it was easier to find information. For example, when considering the auto-ecology of the invasive decapods, they are favored by unstable and warmer environments, which potentially increase their impacts on the native biodiversity and ecosystem services. The same consideration is valid for the three invasive bivalves (*D. polymorpha*, *C. fluminea* and *S. woodiana*) that were classified by the AS-ISK as having a high level of invasiveness. Despite bivalves being more prone to being affected by extreme events (e.g., droughts) compared to decapods, which are more mobile, their high adaptability and resilience allow them to survive, even in challenging environmental conditions. On the contrary, similar to native species, some IAS could even be negatively affected by climate change.

Another important issue to consider when applying AS-ISK is the definition of the threshold values for the BRA and BRA + CCA that the user has to impose on the software. In this paper, we decided to use the same thresholds proposed by Paganelli et al. [18] because of the similarity between the two RA areas.

However, a recent paper on the application of the AS-ISK on a global scale [47] proposed lower thresholds (BRA 13.25 and BRA + CCA 25.75) than the ones used here. The authors suggested classifying alien species with a BRA score above 30 as being "high

risk" and those with a BRA + CCA score above 36 as being "very high risk". Using these thresholds, almost all of the alien species in this study should be classified as having a "medium risk" or "high/very high risk" of invasion, but according to us, this is not always the case (Table 3).

On the other hand, it is also true that a change in the threshold values could lead to a different invasiveness level, but nothing can change the effect of the ecological and economic impacts that these species cause.

## 5. Conclusions

Invasive species are driven by human activities, so solutions to these problems are only possible with a change in human behavior. The management of IAS and their pathways of introduction are the main core of EU regulation 1143/2014. The species included in the List of Union Concern are subject to measures that include restrictions on keeping, importing, selling, breeding, and growing. Member States are required to act on pathways of unintentional introduction, to take measures for the early detection and rapid eradication of these species, and to manage species that are already widespread in their territory. In an area with several anthropogenic pressures (i.e., agriculture, industry, tourism) such as the hydrographic system of Lake Maggiore, the "do-nothing-option", often claimed as the most natural approach to test the ability of ecosystems to recover, will not work.

The management of IAS starts from a general screening of their invasiveness, and the risk assessment tool we used in this paper allowed us to identify the impacts of 16 invasive alien species in the risk assessment area. Moreover, the results of the assessment tool were in accordance with the preventive analysis, classifying seven of the eight IAS indicated by the preventive analysis as having a "high risk" of invasiveness.

Most of the alien species assessed in this study are not included in the List of Union Concern, but this does not mean that they do not have any impacts. If we consider that most of the assessed alien species show bio-ecological traits that are typical of successful invaders [83] and that they often reach high densities in the RA area, they could represent perfect candidates for the next update of the EU List of Union Concern. Nevertheless, it is also true that for most of these IAS, it is very difficult to perform any actions to eradicate them because they are already too abundant, widespread, and well-established in the assessment area; however, it is still important to be aware of the impacts that they can cause.

Over the last few years, some attempts to eradicate aquatic IAS have been made, but no techniques exist to reduce their population without killing other organisms. Nevertheless, when the opportunity to eradicate an IAS is not feasible, the only option is to attempt to control them.

The performed risk assessment allowed us to better understand which macroinvertebrate IAS managers and researchers should focus their actions of surveillance and monitoring programs on, with the aim of tracing their movements and containing their spread.

In our case study, considering that the assessment area is divided between Italy and Switzerland, it would be beneficial to have a joint legislation shared between these two countries to regulate the anthropogenic activities that are responsible for the introduction of IAS.

Furthermore, joint feasible actions are recommended to reduce the potential impacts of macroinvertebrate IAS: rather than focusing on eradication actions only, it may be advisable to improve the quality of the environment through restoration/mitigation actions, thus favoring native species' competition against IAS.

Finally, increasing citizen awareness has become pivotal, and using scientific and correct information to promote good practices to contrast the spread of IAS and to prevent new introductions is necessary.

In 2019, Lombardy Region published the guidelines for the management of alien crayfish species, where they suggest several actions and good practices to mitigate the impacts of these invasive crayfish [84] (www.naturachevale.it, accessed on 11 March 2021)

This could represent a good example of experience exchange that could be proposed in other regions or countries that are close by, such as Piedmont and Switzerland.

In conclusion, this paper represents a first step in the assessment of the macroinvertebrate IAS in the hydrographic system of Lake Maggiore, and it could help to refine potential management strategies that are aimed at reducing the impacts of these species.

**Author Contributions:** Conceptualization, D.P. and A.B.; Data curation, D.P.; Formal analysis, D.P.; Investigation, D.P.; Methodology, D.P.; Resources, A.B.; Software, D.P.; Supervision, L.K. and A.B.; Validation, L.K., L.G. and A.B.; Visualization, D.P. and S.Z.; Writing—original draft, D.P. and A.B.; Writing—review & editing, D.P., L.K., S.Z., L.G. and A.B. All authors have read and agreed to the published version of the manuscript.

**Funding:** No funding were received to carry out this research.

**Institutional Review Board Statement:** Not applicable.

**Informed Consent Statement:** Not applicable.

**Data Availability Statement:** Not applicable.

**Acknowledgments:** We are particularly grateful to two anonymous reviewers for their very valuable comments and suggestions influencing the final version of the paper.

**Conflicts of Interest:** The authors declare no conflict of interest.

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
