# Peer review of "Impacts Analysis of Alien Macroinvertebrate Species in the Hydrographic System of a Subalpine Lake on the Italian–Swiss Border"

_water, doi:10.3390/w13213146_

Round 1

Reviewer 1 Report

This study may contribute to the management of nonnative macroinvertebrates in the Lake Maggiore drainage area. The manuscript is generally well written. Most of my comments are largely minor.

  1. Threshold values of BRA and BRA+CCA

As the authors acknowledge in the Discussion, the interpretation of invasiveness largely depends on the threshold values of BRA and BRA+CCA. Rather than using only the threshold values used in Paganelli et al (2018), the result of invasiveness may be better shown under multiple threshold values or scenarios. The authors mention about Vilizzi et al (2021)’s study on global scale screening on nonnative aquatic organisms. Perhaps authors may also consider using intermediate threshold values, if use of such threshold values produce convincing results. A multiple scenario approach has been adopted in risk analysis of crayfish (Tricarico et al. 2010, Risk Analysis). The results from multiple scenarios may be better shown in the Table.

E Tricarico, L Vilizzi, F Gherardi, GH Copp (2010) Calibration of FI‐ISK, an invasiveness screening tool for nonnative freshwater invertebrates. Risk Analysis: 30: 285-292

  1. Title, L10 and throughout the text “--- macroinvertebrate alien species ---"

Change to “alien macroinvertebrates” or “alien macroinvertebrate species”.

  1. L129 and throughout “IAS”

The present study includes potentially invasive species as well as noninvasive species. At least the three amphipods have no records of invasiveness in any aspect. In this case, the term invasive alien species (IAS) may not be applicable to all screened nonnative species. I would recommend using “alien species” or “nonnative species” to tables, figures and in the text wherever the author mention about the entire nonnative species screened in the study.

  1. L19 Orconectes (Faxonius)

Use of new genera name in parenthesis is confusing.

Faxonius looks like the subgenera name.

Instead, it would be better to state “Faxonius limosus (formerly Orconectes limosus)in the abstract.

Afterwards, you can just spell as “Faxonius limosus” throughout the paper.

  1. Figure 1

What do the dashed lines indicate? Are they political boundaries? It may be better to show province and national boundaries in different lines and include explanations in the legend or in the figure.

  1. Table 1

It would be easier to understand if you include common names of each nonnative species.

  1. Figure 2 and other Figures

Orconectes limosus should be changed to Faxonius limosus.

  1. Figure 2 legend “--- indicate the and the Basic Risk Assessment”

Omit “the and”.

  1. L271 inhibition of primary production

This phrase needs to be corrected.

This statement is not accurate nor true. Rodriguez et al (2003) reported that macrophyte grazing or destruction by Procambarus clarkii caused stable state changes from a macrophyte-dominated clear water state to a phytoplankton-dominated turbid state in a shallow lake. In such case, primary productivity was probably more enhanced after crayfish invasion via outbreak of phytoplankton and cyanobacteria, although they did not measure primary productivity in their study.

  1. L267-274

The greatest threats of crayfish invasion lie in their roles as carriers of diseases such as Aphanomyces astaci and white spot syndrome virus. Specifically, American crayfishes serve as carriers of the crayfish plague fungus, Aphanomyces astaci, which is listed as one of IUCN’s 100 of the worst invasive alien species. The carriers of the fungus include various American crayfishes and are not limited to Procambarus clarkii (Svoboda et al 2017). The statement in L286-288 should be moved to the earlier part of L267-274. The information that many American crayfishes have potential to serve as carriers of Aphanomyces astaci may need to be added in the sentence. This is not general characteristics of all invasive crayfishes, as non-American (incl. European) crayfishes are highly vulnerable to the fungus (Unestam 1969, Svoboda et al 2017).

Svoboda, J., Mrugała, A., Kozubíková-Balcarová, E., Petrusek, A. (2017) Hosts and transmission of the crayfish plague pathogen Aphanomyces astaci: a review. Journal of Fish Diseases 40(1), pp. 127-140

Unestam T. (1969) Resistance to the crayfish plague in some American, Japanese and European crayfishes. Report of the Institute of Freshwater Research, Drottningholm 49, 202–209.

  1. L261-263

Is Pontastacus leptodactylus missing from here?

  1. L316 100 worst invasive species in the world

The correct term is “100 of the world’s worst invasive alien species”

There is a citation for this booklet.

See:

  1. L 325 aggressive

I am not certain whether “aggressive” is proper in this sentence.

It sounds like Dreissena polymorpha attacks other animals through interference.

May be better to use other term such as successful (?).

Author Response

point-by-point response letter

Reviewer #1

  1. Threshold values of BRA and BRA+CCA

As the authors acknowledge in the Discussion, the interpretation of invasiveness largely depends on the threshold values of BRA and BRA+CCA. Rather than using only the threshold values used in Paganelli et al (2018), the result of invasiveness may be better shown under multiple threshold values or scenarios. The authors mention about Vilizzi et al (2021)’s study on global scale screening on nonnative aquatic organisms. Perhaps authors may also consider using intermediate threshold values, if use of such threshold values produce convincing results. A multiple scenario approach has been adopted in risk analysis of crayfish (Tricarico et al. 2010, Risk Analysis). The results from multiple scenarios may be better shown in the Table.

E Tricarico, L Vilizzi, F Gherardi, GH Copp (2010) Calibration of FI‐ISK, an invasiveness screening tool for nonnative freshwater invertebrates. Risk Analysis: 30: 285-292

Reply: Thank you for this suggestion. We have added the two different scenarios to table 3 and modified the caption

  1. Title, L10 and throughout the text “--- macroinvertebrate alien species ---" Change to “alien macroinvertebrates” or “alien macroinvertebrate species”.

Reply: Thank you for notice that. We have modified the title and the way of saying according to your suggestion

  1. L129 and throughout “IAS”

The present study includes potentially invasive species as well as noninvasive species. At least the three amphipods have no records of invasiveness in any aspect. In this case, the term invasive alien species (IAS) may not be applicable to all screened nonnative species. I would recommend using “alien species” or “nonnative species” to tables, figures and in the text wherever the author mention about the entire nonnative species screened in the study.

Reply: Thank you, we have modified the term IAS with alien species in the text, figures captions and table as you suggested

  1. L19 Orconectes (Faxonius)

Use of new genera name in parenthesis is confusing. Faxonius looks like the subgenera name. Instead, it would be better to state “Faxonius limosus (formerly Orconectes limosus)in the abstract. Afterwards, you can just spell as “Faxonius limosus” throughout the paper.

Reply: Thank you for this comment. We have replaced the old name with the new one and we have modified the abstract

  1. Figure 1

What do the dashed lines indicate? Are they political boundaries? It may be better to show province and national boundaries in different lines and include explanations in the legend or in the figure.

Reply: We modified the figure 1 and its caption specifying the national and regional borders

  1. Table 1

It would be easier to understand if you include common names of each nonnative species.

Reply: When it was possible, we have added the common name of the alien species

  1. Figure 2 and other Figures

Orconectes limosus should be changed to Faxonius limosus.

Reply: modified as suggested

  1. Figure 2 legend “--- indicate the and the Basic Risk Assessment”

Omit “the and”.

Reply: modified as suggested

  1. L271 inhibition of primary production

This phrase needs to be corrected. This statement is not accurate nor true. Rodriguez et al (2003)

reported that macrophyte grazing or destruction by Procambarus clarkii caused stable state changes from a macrophyte dominated clear water state to a phytoplankton-dominated turbid state in a shallow lake. In such case, primary productivity was probably more enhanced after crayfish invasion via outbreak of phytoplankton and cyanobacteria, although they did not measure primary productivity in their study.

Reply: Thank you for this comment. We have modified the sentence according to your suggestion (line 275-276)

  1. L267-274

The greatest threats of crayfish invasion lie in their roles as carriers of diseases such as Aphanomyces astaci and white spot syndrome virus. Specifically, American crayfishes serve as carriers of the crayfish plague fungus, Aphanomyces astaci, which is listed as one of IUCN’s 100 of the worst invasive alien species. The carriers of the fungus include various American crayfishes and are not limited to Procambarus clarkii (Svoboda et al 2017). The statement in L286-288 should be moved to the earlier part of L267-274. The information that many American crayfishes have potential to serve as carriers of Aphanomyces astaci may need to be added in the sentence. This is not general characteristics of all invasive crayfishes, as non-American (incl. European) crayfishes are highly vulnerable to the fungus (Unestam 1969, Svoboda et al 2017).

Svoboda, J., Mrugała, A., Kozubíková-Balcarová, E., Petrusek, A. (2017) Hosts and transmission of the crayfish plague pathogen Aphanomyces astaci: a review. Journal of Fish Diseases 40(1), pp. 127-140

Unestam T. (1969) Resistance to the crayfish plague in some American, Japanese and European crayfishes. Report of the Institute of Freshwater Research, Drottningholm 49, 202–209.

Reply: thank you for these suggestions. We have moved and modified the sentence (lines 279-282). We have also added the more recent reference (Svoboda et al., 2017) to our list.

  1. L261-263

Is Pontastacus leptodactylus missing from here?

Reply: No, it is not missing. Here, we described the results of the “preventive analysis”. According to this analysis, P. leptodactylus is not considered so dangerous because it is not a threat to human beings or to the native biodiversity.

  1. L316

100 worst invasive species in the world The correct term is “100 of the world’s worst invasive alien species” There is a citation for this booklet.

See:

Reply: Thanks, we have modified this citation

  1. L 325 aggressive

I am not certain whether “aggressive” is proper in this sentence. It sounds like Dreissena polymorpha attacks other animals through interference. May be better to use other term such as successful (?).

Reply: thank you, we have modified the term as you suggested (line 335)

Reviewer 2 Report

The manuscript covers an interesting topic, the risk assessment of invasive alien species in Lago Maggiore, which is interesting and useful also in a more general sense and wider application. I have only minor comments in the attached pdf, mostly concerning language clarity and use of English. The English is generally well understandable, but the manuscript could benefit from being read and corrected by a native speaker.

Author Response

point-by-point response letter

Reviewer #2

The manuscript covers an interesting topic, the risk assessment of invasive alien species in Lago Maggiore, which is interesting and useful also in a more general sense and wider application. I have only minor comments in the attached pdf, mostly concerning language clarity and use of English. The English is generally well understandable, but the manuscript could benefit from being read and corrected by a native speaker.

Reply: Thank you for all your comments in the pdf file. We have modified the sentences as you suggested, and we have asked to a native English speaker to check the manuscript. We have highlighted all changes in the manuscript file through track changes.